# 'Treading water but drowning slowly': what are GPs' experiences of living and working with mental illness and distress in England? A qualitative study

Ruth Riley,[1] Johanna Spiers,[2] Carolyn A Chew-Graham,[3] Anna K Taylor,[4] Gail A Thornton,[2] Marta Buszewicz[5]

[1]Institute of Applied Health Research, University of Birmingham, Birmingham, UK
[2]Centre for Academic Primary Care, University of Bristol, Bristol, UK
[3]Research Institute, Primary Care and Health Sciences, Keele University, Keele, UK
[4]Faculty of Health Sciences, University of Bristol, Bristol, UK
[5]Research Department of Primary Care and Population Health, University College London, London, UK

**Correspondence to**
Dr Ruth Riley;
r.riley@bham.ac.uk

## ABSTRACT

**Objectives** This paper provides an in-depth account of general practitioners' (GPs) experiences of living and working with mental illness and distress, as part of a wider study reporting the barriers and facilitators to help-seeking for mental illness and burn-out, and sources of stress/distress for GP participants.

**Design** Qualitative study using in-depth interviews with 47 GP participants. The interviews were audio recorded, transcribed, anonymised and imported into NVivo V.11 to facilitate data management. Data were analysed using a thematic analysis employing the constant comparative method.

**Setting** England.

**Participants** A purposive sample of GP participants who self-identified as: (1) currently living with mental distress, (2) returning to work following treatment, (3) off sick or retired early as a result of mental distress or (4) without experience of mental distress. Interviews were conducted face to face or over the telephone.

**Results** The findings report GP participants' in-depth experiences of distress and mental illness with many recollecting their distressing experiences and significant psychological and physical symptoms relating to chronic stress, anxiety, depression and/or burn-out, and a quarter articulating thoughts of suicide. Many talked about their shame, humiliation and embarrassment at their perceived inability to cope with the stresses of their job and/or their symptoms of mental illness.

**Conclusions** These findings paint a concerning picture of the situation affecting primary care doctors, with participants' accounts suggesting there is a considerable degree of mental ill health and reduced well-being among GPs. The solutions are complex and lie in prevention and provision. There needs to be greater recognition of the components and cumulative effect of occupational stressors for doctors, such as the increasing workload and the clinical and emotional demands of the job, as well as the need for a culture shift within medicine to more supportive and compassionate work environments.

### Strengths and limitations of this study

► Few studies employing qualitative methods, using in-depth interviews, have been used to examine this topic and depict the experience of distress among general practitioners (GPs).

► This study carried out 47 interviews with GPs from across England and contributes to a growing body of research illuminating and examining the experience of living with chronic stress and distress among this population of doctors.

► Participants were self-selecting, which may be perceived as a limitation; however, the recruitment of participants ensured that the sample was varied in terms of age, gender, number of years as practising GPs, level of seniority/employment status in the practice and geographical location.

► This variation provided a range of perspectives; however, uniform differences between the groups were not observed.

► The researchers did not employ respondent validation; however, the second coders included academic GPs and team members with lived experience of mental illness which afforded some checks and balances to the validity of the analytical process, interpretation of data and generalisability of the research findings.

► In recruiting individuals with experience of mental illness and burn-out, we also included the perspectives of GPs who had no disclosed history of mental illness which enables the data to be more widely transferable to the UK GP population.

► The multidisciplinary research team independently analysed a subset of transcripts in order to contribute to the generation and refinement of codes to maximise rigour.

► Emergent themes were subsequently discussed by the whole team to ensure credibility and confirmability.

## INTRODUCTION

Compared with the general population, doctors, including general practitioners (GPs) working in primary care, experience high levels of psychological distress linked to chronic stress, anxiety, depression and burn-out.[1–4] These doctors also have higher rates of mental illness and suicide.[5–8]

Evidence from industrialised countries (eg, UK, USA, Australia) indicates that suicide rates are higher among GPs, psychiatrists and anaesthetists compared with other medical specialties.[6 8] Doctors undergoing fitness for practice investigations are at particularly high risk of suicide.[9] Compared with the general population, doctors also have higher rates of drug and alcohol misuse and dependency, with a higher rate of misuse of prescription medicines.[10] Psychological difficulties or distress experienced by doctors can be further compounded by self and societal stigmatising associated with mental illness, with evidence that doctors avoid or delay disclosure and help-seeking due to concerns about loss of confidentiality, denial of illness and logistical barriers to accessing support.[11–15] The prevailing hegemonic culture among medical practitioners and non-disclosure of vulnerability, with shame attached to mental health problems or psychological difficulties and the consequent perceived inability to cope within the job, also hinders help-seeking.[4 11 16 17] It has been suggested that personality characteristics such as perfectionism and obsessionality, while being positively associated with hard-working and conscientious professionals, may play a role in predisposing doctors towards chronic stress and/or mental ill health.[4]

The increasing prevalence of mental ill health among the medical profession, including GPs, has been attributed to a number of factors, most notably the escalating occupational stresses related to workload, long hours and the emotional demands of the job.[4 18–23] In addition, sharp increases in patient demand[19 24] and transfer of care from specialist to primary care,[25] as well as reductions in resources,[25] have resulted in the pressure on primary care in England being at its highest ever. This pressure is predicted to increase in the future.[25] Findings such as these highlight the potentially stressful context in which GPs are currently working, leaving many vulnerable to distress and/or mental health challenges.[4 13 14 23]

Evidence of the extent of mental ill health and burn-out among GPs is largely derived from survey data, with some small-scale qualitative studies exploring the topic.[13 26–30] To date, few studies have explored the experiences and accounts of GPs living and working with enduring levels of stress, mental ill health and psychological distress using in-depth qualitative methods. The employment of in-depth interviews in this study allowed GP participants to describe the full extent and complexity of their distress. In addition, by providing the 'patient's view',[31] qualitative findings can fully elucidate details and context about phenomena in a way that quantitative work cannot.[32] This paper reports GPs' detailed experiences of living and working with mental illness and emotional distress, as part of a wider study reporting the sources of stress and distress[23] for GPs and the barriers and facilitators to help-seeking for mental illness and burn-out[14] among this medical population.

## METHODS

This multicentre, qualitative study employed in-depth interviews with 47 GPs in England. Information about the study was advertised through professional publications such as Pulse, study newsletters, social media and national and local GP networks (including local medical committees) in Bristol, Manchester and London. A subsample (n=12) was recruited through a specialist treatment service. GPs interested in taking part were asked to self-select into the following groups: (1) Living with anxiety, depression, stress and/or, burn-out; (2) Returning to work following treatment; (3) Off sick or retired early due to psychological ill health (4) No mental ill health. GPs were offered reimbursement of £80 to recognise the time for their participation.

GPs who expressed an interest in participating were purposively sampled to represent as even a spread as possible across these four groups, although the largest number of participants were in group 1. We intended to recruit approximately 10 participants per group. However, the majority of GPs who contacted us self-selected into group 1 and, due to the emergent rich data, continued recruitment to this group and further exploration of emerging themes was justified in meeting the study's aims and objectives. We endeavoured to recruit more participants into groups 2 and 4 using targeted publicity information, but because of time constraints those groups remained marginally under-recruited. In the event, many GPs who identified as living with no stress reported as having had experiences of stress and distress at some juncture in their career. More female GPs contacted the study team expressing an interest in participating, and therefore the disparity in numbers interviewed reflects this. The iterative process of recruitment, sampling and analysis ensured that emerging concepts and themes could be tested out among participants with different characteristics (eg, partners vs locum GPs).

Face-to-face or telephone interviews lasting between 27 and 126 min (mean=69 min) were conducted between April and September 2016. The majority of interviews were conducted in participants' homes. The in-depth interviews were conducted by two authors (JS, RR) using a flexible topic guide (see box 1). This was informed by the existing literature, input from GPs on the study team and PPI consultation exercises conducted with GP networks prior to obtaining funding. The interviews were audio recorded, transcribed, anonymised and imported into NVivo V.11 to facilitate data management. Analysis and data collection were conducted iteratively. A thematic analysis was performed involving a process of constant comparison between cases.[33] Analysis commenced with JS generating an initial coding framework grounded in the data, which was added to and refined, with material regrouped and recoded as new data were gathered. Codes were gradually built into broader categories through comparison across transcripts, and higher-level recurring themes were developed until data saturation was reached when no new themes were arising from the data.[34]

## Box 1   Interview topic guide

**Introduction and background**
► Describe average working day in practice (hours, surgery, home visits) and any additional responsibilities.

**Current well-being**
► Explore current well-being, feelings about work, levels of stress, work– life balance.
► Explore causes of stress/distress (workload, hours, admin, clinical caseloads, organisational issues, lack of support, personal issues, pre-existing mental health symptoms).
► Explore reasons for early retirement/sickness (if relevant).

**Managing stress**
► Explore how general practitioners (GPs) manage their workload/stress in their day-to-day work life, what they do to relax, to look after themselves (self-care strategies: supportive relationships, sport, exercise relaxation techniques).
► Explore relationship with colleagues and whether/how/if concerns are raised, how they are responded.
► Explore if GPs have received informal/formal supervision or mentor (1:1 or group) and experience/value of group.
► Explore thoughts/feelings about seeking help, barriers to seeking help (stigma/shame, fears about confidentiality, uncertainty of where to go).

Reflexivity was employed throughout the research. Both interviewers were experienced qualitative researchers who both reflected on and discussed the impact of the data on their cognitive and emotional sensing throughout the study. Both researchers also discussed and made explicit how their epistemological (JS with a background in psychology and RR in medical sociology) and experiential backgrounds may have oriented the data collection and analytical process.

Members of the multidisciplinary research team independently analysed a subset of transcripts in order to contribute to the generation and refinement of codes to maximise rigour. Emergent themes were discussed by the whole team to ensure credibility and confirmability.

## RESULTS

Willing participants were purposively sampled from GPs who expressed an interest in the study, based on demographic and practice characteristics as well as self-selected group type. The characteristics of the participants are found in table 1.

For this paper, we will present the following five main themes and corresponding subthemes relating to:
1. Burn-out—loss of empathy or compassion fatigue, feeling empty and/or exhausted, difficulty making decisions, low job satisfaction and loss of enjoyment in seeing patients.
2. Anxiety—psychological symptoms relating to anxiety, including feeling anxious or on edge, worrying, irritability, inability to relax and physical symptoms related to anxiety such as panic attacks, hyperventilating, palpitations, sweating and nausea.

3. Depression—distressed at work, insomnia and exhaustion, negative thoughts, low self-esteem, irritability and anger.
4. Suicidal ideation—thoughts of suicide.
5. Shame, as well as feelings of failure and humiliation, resulting from a perceived lack of resilience and inability to cope.

It should be noted that some participants described or disclosed symptoms that could be clearly attributed to chronic stress, anxiety, depression and burn-out. Other participants described or framed their symptoms in terms of diagnostic concepts associated with these categories, while others talked about their feelings and emotional state without necessarily labelling them. Notably, the symptomatology was identified across all groups including group 4, that is, individuals who self-identified as having no mental health concerns. The themes also applied to

**Table 1** Participant and practice characteristics

| GP characteristics | n=47 |
|---|---|
| | N |
| Sex (female) | 33 |
| Age | |
| 20–29 | 1 |
| 30–39 | 12 |
| 40–49 | 20 |
| 50–59 | 14 |
| Group | |
| 1 | 19 |
| 2 | 9 |
| 3 | 11 |
| 4 | 8 |
| Years since qualified | |
| <10 | 19 |
| No of sessions contracted per week | |
| <5 sessions (mean actual hours worked) | 12 (15) |
| >5 sessions (mean actual hours worked) | 32 (38) |
| Fully retired | 3 |
| Mean size of practice | 12 624 |
| Range | 3600–37 000 |
| Employment status | |
| Partner | 17 |
| Salaried | 11 |
| Locum | 5 |
| Registrar | 4 |
| Retired | 3 |
| Sick leave | 5 |
| More than one role | 2 |

GP, general practitioner.

participants irrespective of seniority or role in the practice, age or gender. There was no clear distinction in terms of presentation or severity of symptoms between the groups and so no typology could be developed along these lines.

## Symptoms of burn-out

Some participants reported feeling a loss of empathy or compassion fatigue, feeling empty, exhaustion, difficulty making decisions, low job satisfaction and loss of enjoyment for their work, using language which highlights the depth of the feelings:

> I'd lost my empathy with my patients. You know, and er I'm just—I was just—you're just like a ketchup bottle in a production line […] But from a personal point of view I'm still feeling a bit washed out by it, you know, I feel a little bit empty. You know, I feel like I'm burnt out really a bit, to be honest. (P25, Male Salaried)

> I was saying all the right things but […] I felt quite detached from it. (P27, Female Salaried)

The following participant describes how feeling overworked can lead to a reduced ability to offer compassion and support to patients:

> If you find yourself working too many sessions, I can, you know, I feel it, and you almost feel like you lose the milk of human kindness. And you can't do that job if you're not (pause) at your best. (P21, Female, Salaried and Locum)

## Symptoms of anxiety

The majority of participants, including those who self-identified as having no mental health concerns (group 4), described psychological symptoms relating to anxiety, including feeling on edge or hyperalert, persistent worrying, irritability and an inability to relax. The following examples demonstrate how difficult and distressing these symptoms and feelings were for some participants to negotiate:

> Everything was running around in my mind at 200 miles an hour. I was in sort of hyper drive and, yeah, you wake up to super stimulated, super awake, super, you know, on edge and anxious. (P36, Male Partner)

Many participants also described physical symptoms related to their anxiety, such as panic attacks, hyperventilating, palpitations, sweating and nausea:

> Erm (pause) and I, and I was getting quite bad chest pain, palpitations, I was feeling anxious in many circumstances, like the school run, just meeting people, going to meetings and it was becoming incapacitating where it felt like somebody had got a knife in my chest all the time. And I knew it wasn't anything to do with my heart. (P31, Female Locum)

## Symptoms of depression

Many GP participants described symptoms relating to depression, which included: feeling distressed at work, low mood, hopelessness, sleep problems, low self-esteem, guilt, restlessness, fatigue and suicidal feelings. Some participants employed a range of poignant and powerful metaphors which illustrate the extent of those difficult emotions:

> And, to be honest with you, that kind of er—the best way that I could describe that is that it's kind of like a dark shadow that's in the corner of the room […] I basically felt like I was treading water but drowning slowly. (P45, Male Salaried)

### Distressed at work

Many participants recalled their experiences of feeling distressed, anxious and tearful, with some describing breaking down at home, after or during work or in front of patients:

> I mean there were times certainly when I arrived at work and just sat outside and cried before I even went in the building. And I often cried on the way home. I'd often cry when I came in from work, um just collapse in a heap, and get to bed actually crying. (P31, Female Locum)

> I remember once actually sitting under the desk, hugging my knees to my chest, I was so anxious. (P25, Male Salaried)

> I would just end up in tears all the time at the end of each session. (P53, Female Salaried)

### Insomnia and exhaustion

Among the myriad of symptoms, insomnia and exhaustion were a commonplace experience for study participants:

> Um - so I wasn't sleeping and I wasn't—wasn't eating very well, and—and it just all sort of took its toll. (P23, Female Salaried)

> It's very hard to function when you're not ever getting refreshing sleep […] And I just wouldn't stop. And absolute, incapacitating terror. Um—So, but one's having um—night, night terrors maybe two, three, four times a week. Sometimes more than one a night. (P24, Female Partner)

Many GP participants detailed their exhaustion, which was often linked to insomnia and could be symptomatic of their anxiety, depression and/or feeling burnt-out, demonstrating the ways in which different elements of GP experience combine to add to distress:

> Probably like sort of swimming through treacle, so everything seemed—um—much harder. (P38, Male Partner)

> So like you can't, you can't avoid that fatigue. You might be able to put it off a bit (laughs) but you're going to pay interest on it. (P7, Female Locum)

### Negative thoughts/low self-esteem

Some participants talked about grappling with cycles of negative thoughts, symptomatic of low self-esteem and a common symptom of depression, as this participant illustrates:

> And so then I'm really struggling, because I'm struggling with my internal dialogue saying, you know, 'Not good enough, could be doing a better job. Not really, not really er making people better.' […] persistent negative cycles of er (pause), yeah, of negative thoughts about yourself (P20, Male Partner)

### Irritability and anger

Some study participants described feeling irritable and sometimes angry; a possible symptom of depression in this context:

> But I think, towards the end when I stopped locum-ing again, I was beginning to feel very tired as a result and that it was taking more out of me than it ought to really, and feeling um, yeah, feeling more irritable. (P47, Female on sick leave)

This account demonstrates how feeling stressed and irritable can sometimes manifest itself within a work context:

> …it was fairly um traumatic really, I have to say… I think I'd become quite stressed at work actually and I'd lost my temper in meetings at times and um nobody quite knew what to do about it. (P39, Male Partner)

### Suicidal ideation

One in four GP participants described having had suicidal thoughts, with the following participants reflecting on occasions when they had contemplated and ruminated on ways of taking their own lives:

> And it just came on me like a wave. And then I started thinking, 'I'm going to jump onto those train tracks.' (P45, Male Salaried)

> And um constant ruminations on suicide and doing myself in. And the trick with suicide is to persuade yourself that those around you would be better off without you, um and I got to that point and I decided on ways that I might (pause) do myself in and ruminated about that all the time. Just, it just filled my waking and sleeping nights. (P20, Male Partner)

This latter data extract demonstrates how suicidal thoughts can be both constant and normalised for some GPs.

The following GP reflects on the advice she gives to her patients who express suicidal thoughts and how her feelings mirrored her patients' expressed hopelessness and despair, again showing the ways in which GP experiences can combine to add to distress:

> Um, and then I think the hard thing, the worst thing I think I found was when people were feeling suicidal, and they were saying they were feeling suicidal and why they were feeling suicidal. And normally I'd be like, 'Look, you know, it's temporary, it'll get better. These are the things that we can do,' but I just sat there thinking, 'Well, you're probably right, it is rubbish,' you know. (P27, Female Partner)

### Shame and feelings of failure

Perhaps symptomatic of the stigma attached to mental illness and the culture of invulnerability within medicine, some GP participants described how they felt ashamed, embarrassed, humiliated and a sense of having failed, due to their perceived lack of resilience and inability to cope. These quotations illustrate the depth of that emotion for two of the participants:

> And you do feel a huge sense of shame, huge sense of shame. Because even though I treat patients day in, day out with anxiety and depression, it felt like this was my not coping. And I was thinking, 'Why are the partners all able to cope with this but not me?' and that sort of thing. So it was all rolled into one really. (P4, Female Partner)

> Er (sigh) yeah, I think for me it was very embarrassing. Didn't want to admit it. Couldn't believe that like I'd got to this stage. Cos I'd never been someone like that […] yeah, I did find it really hard, I think, to actually accept that this had happened. Um (pause) yeah, bit of a failure? I suppose, in a way. Kind of feel like you're not strong enough? (P44, Female Registrar)

The participant below highlights how the denial and intolerance of mental ill health among some colleagues can enter into an individual's internal dialogue, thus compounding existing feelings of shame:

> 'Yeah, well you don't get ill, do you? You know, you, you refuse to accept any illness in yourself, don't you? It's just a question of that's why doctors present late with illnesses because they don't like to make a fuss. We see it as a sign of weakness, which I think probably is why we are intolerant of illness in our partners, because we're doctors. And I think it's just part of the sort of macho thing which is first drummed into you at medical school and the end product of that generation, which—if you can't take the heat get out of the kitchen, type thing'. (P49, Female, Retired)

## DISCUSSION

The accounts of the 47 participants in this study suggest there may be an undetermined population of GPs who are living and working with poor mental health. Participants vividly described and recollected their distressing experiences and troubling symptoms, with one in four GPs articulating thoughts of suicide. Such findings are

currently under-reported in the literature with few studies providing evidence of suicidal thoughts and extreme levels of distress among this GP population. The GP participants described numerous troubling psychological and physical symptoms relating to chronic stress, anxiety, depression and/or burn-out with several breaking down at work—crying, feeling despair and some contemplating suicide. Many also talked about their shame, humiliation and embarrassment at their perceived inability to cope with the stresses of their job and/or their symptoms of mental ill health. The experiences and feelings of many GP participants depict a dispiriting emotional landscape, with notable signs of poor mental health and striking levels of distress.

There has been a paucity of in-depth qualitative research exploring this topic area with doctors, including GPs.[14] This study highlights that many GPs are living and working with extreme levels of emotional distress. A notable proportion of participants vividly recalled and articulated both fleeting and recurring thoughts of suicide, which is reflective of the extent of individuals' distress and suffering. These candid, evocative and sometimes disturbing accounts, some hitherto unvoiced, reveal a population of doctors with generally very poor mental health.

The findings from this study and the related papers[14 23] were based on in-depth interviews with 47 GPs from across England and contribute to a growing corpus of research illuminating and examining the causes, experiences and impact of chronic stress and distress among GPs.[20 35] The participants were self-selecting, which may be seen as a limitation; however, the recruitment of participants ensured that the sample was varied in terms of age, gender, number of years as practising GPs, level of seniority/employment status in the practice and geographical location. In addition to recruiting individuals with self-reported mental ill health and burn-out, we also included the perspectives of GPs who had no disclosed history of mental health problems, which enables the data to be more transferable to the wider GP population. However, despite this heterogeneity within the sample, we did not observe any marked differences between the groups, suggesting a certain uniformity of distress for the group despite demographic differences.

These findings present a concerning picture of psychological difficulties and reduced well-being affecting primary care doctors. The experience of chronic stress and anxiety in GPs (eg, feeling tense, restless, anxious, worried or having disrupted sleep, panic attacks, hyperventilating, palpitations, sweating and nausea) appears commonplace, including in those who did not identify themselves as having mental ill health. Our findings reflect the evidence that doctors frequently normalise, minimise, deflect or downplay how stressed they feel, which may preclude help-seeking and the availability of appropriate support.[36] Strikingly, one GP participant normalised the way they routinely ruminated on their suicidal thoughts, with other participants indicating that they were living and coping on a day-to-day basis with symptoms of chronic stress, anxiety and depression, ranging from sleeplessness to persistent anxiety. Evidence indicates that chronic stress negatively impacts on physical health and is a key risk factor for mental illness, including anxiety and depression, and can lead to the adoption of maladaptive coping mechanisms such as nicotine, alcohol, and illicit drug use.[37]

Normalising and acknowledging chronic stress as an occupational hazard associated with being a doctor is helpful, but only if solutions are sought which focus on prevention and the provision of support to maintain the health and well-being of frontline clinicians. Since January 2017, GPs and GP trainees in England suffering from mental ill health and addiction have been able to access free support through the National Health Service's GP Health Service.[38] Further research and evaluation must be undertaken to assess the effectiveness and accessibility of this service. The emphasis now needs to focus on preventing mental ill health and improving the support available to doctors. This findings highlight that internalised and perceived stigma still prevent doctors from seeking support.[11]

Access to personal, relational and environmental resources are protective factors for good mental health and well-being and impact on individuals' resilience.[39] Crucially, collegial support is a protective factor for good mental health—support from mentors, supervisors and colleagues is associated with resilience and reduced sickness.[22]

Balint groups or similarly structured group work or supervision continue to be employed in general practice and are valued by GPs,[40] yet are not used across all practices. Individual or group supervision aims to provide a safe and supportive space where staff can openly discuss the pressures and emotional challenges of their work and may, as previous evidence suggests, provide GPs with the support they need while offering protection against compassion fatigue and burn-out.[41] Its wider use may need to be considered.

Tackling the culture of invulnerability early on in medical training is also key. Schwartz Centre Rounds, for instance, are currently being piloted in medical schools with early evidence supporting their value.[42] First,[43] buddy systems, and access to regular supervision or mentorship can also offer support and a reflective space. The British Medical Association has produced guidance on recognising signs of mental distress among colleagues with suggestions about what action to take.[44] Perhaps caution needs to be applied to the over-reliance on personal resilience as a solution as this may shift the responsibility for maintaining good mental health entirely on to the individual. While self-care is paramount in maintaining good mental and physical health, it is not solely the responsibility of individual doctors to look after their mental health; employers and colleagues need to engender a more supportive and compassionate culture. Doctors are not invulnerable to suffering, most especially their own.

In the light of the findings of this study, we recommend the following:

► Greater recognition of the components and cumulative effect of the occupational stressors for doctors, such as increasing workload and the clinical and emotional demands of the job.
► A culture change within medicine, with a shift to more supportive and compassionate work cultures.
► The destigmatising of mental ill health and the recognition of GPs' experiences and emotional responses as symptomatic of occupational stress as opposed to any perceived failure to cope in the individual.

## CONCLUSION

These findings, together with those from the wider study,[14 23] paint a concerning picture of the situation affecting primary care doctors, with participants' accounts suggesting there is a population of GPs experiencing a considerable degree of mental ill health, and overall reduced well-being among GPs. The solutions are complex and prevention lies in addressing the systemic causes of stress (related to diminishing resources, increased workload, etc). In addition to recognising the components and cumulative effect of occupational stressors for doctors, such as the increasing workload and the clinical and emotional demands of the job, there is need for a culture shift within medicine to more supportive and compassionate work environments where it is safe and acceptable for doctors to talk or seek help when they need it. Medical schools, medical institutions and individuals need to address and tackle the stigma of mental illness and engender a greater tolerance and acceptance of vulnerability.

**Acknowledgements** The authors would like to thank all volunteer participants who kindly gave their time to share their experiences of working in English general practice; and collaborators Clare Gerada and Chris Manning for endorsing this study, providing valuable comments on the funding application and assisting with recruitment.

**Collaborators** Clare Gerada; Chris Manning.

**Contributors** RR, JS, CAC-G and MB: substantial contributions to conception and design, acquisition of data or analysis and interpretation of data; drafting the article or revising it critically for important intellectual content and final approval of the version to be published. AKT and GAT: analysis and interpretation of data; drafting the article or revising it critically for important intellectual content; and final approval of the version to be published.

**Funding** The study was funded by NIHR School for Primary Care Research.

**Disclaimer** The views and opinions expressed there in are those of the authors and do not necessarily reflect those of the NIHR School for Primary Care Research, NIHR, NHS or the Department of Health.

**Competing interests** None declared.

**Patient consent** We sought consent from clinicians and doctors as patients.

**Ethics approval** South West—Frenchay Research Ethics Committee (reference number: 15/SW/0350).

**Provenance and peer review** Not commissioned; externally peer reviewed.

**Data sharing statement** This study has not received ethical approval to share confidential data with any third party other than the study research team.

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
