## [Reviewer comments · BMJ Open]

ARTICLE DETAILS

TITLE (PROVISIONAL)	'Treading water but drowning slowly': What are GPs' experiences of living and working with mental illness and distress in England? A qualitative study
AUTHORS	Riley, Ruth; Spiers, Johanna; Chew-Graham, Carolyn; Taylor, Anna; Thornton, Gail; Buszewicz, Marta

VERSION 1 – REVIEW

REVIEWER	Dr Billy Gazard (on behalf of Dr Stephani Hatch) King's College London
REVIEW RETURNED	01-Aug-2017

GENERAL COMMENTS	Recommendation: Major revisions Summary: This paper aims to explore GP experience of living and working with chronic stress, mental ill health and psychological distress using qualitative research methods. The methods are well described and researchers have clearly demonstrated the rigour and validity of the study design and procedures. However, I have concerns about the novelty and implications of the objectives and results of the study given findings already present in the literature that not only describe the extent of psychological distress in doctors and general practitioners but also further describe the relationship between work related risk factors, burn out and psychological distress. The authors point out that few studies have explored these experiences using qualitative research methods and the descriptions that they present are useful in detailing the psychological distress GPs experience. However, I don't think the results or interpretation of the findings in this study build on the quantitative data presented in the research literature. From the presented topic guide, the authors show that they have collected information on both work related stress and managing stress. I would question why they have not incorporated this into the paper especially as the authors allude to these factors in their strengths and limitations section, introduction and discussion. I would recommend major revisions to fully incorporate these into the paper by updating the objectives of the study. Specific suggestions are outlined below: Abstract 1. Page 2 Ln 54-55: In objectives, I think it's confusing for the reader that the wider study is mentioned. I think it's important to mention this in the methods but not sure it's well paced here.2. Page 3 Ln 26-32: In conclusions, I'm not sure that the author can state that there is a considerable degree of mental ill health among GPs due to the self-selecting nature of the sample. I agree with the need for greater recognition of the effect of occupational stress and don't see why this data is not incorporated
---

into the paper if the authors have this data.

Strengths and limitations

1. Page 3 Ln 46: Authors state that the study contributes to body of research on the causes and impact of chronic stress but do not present any findings on the causes of the stress and psychological distress.

2. Page 3 Ln 49-52: authors state a strength of the study is that the sample was varied by sociodemographics but do not go on to state/describe if there were any differences in experiences by key demographics. Are there any differences?

Introduction

1. P4/5 Ln 48-57/ Ln 3-6: The second paragraph of the introduction contains important information on the risk factors for poor mental health in the GP population and how these risk factors are becoming increasingly burdensome. However, I'm not sure why this is presented given the stated objective. It would be useful to see this literature as a basis for a second objective and findings related to this relationship presented in the results section.

2. P5 Ln 7-13: This paragraph states that few qualitative studies have explored GPs experience of working with stress and mental ill health and this seems to be the reason for conducting the analysis. This doesn't convince me as to why the analysis should be conducted given that we know the extent of mental ill health and burnout in GPs using quantitative data. It should be clearer in what this study adds to the literature. Are you aiming to provide more detailed information on the character of stress and mental ill health in GPs than what has been previously reported in the literature? A better argument for the objective of your study would be beneficial.

Methods

1. In Box 1, topic point on describing average working day appears in both introduction and background and current wellbeing. One needs to be deleted.

2. In Box 1, there are several grammatical errors, for example, on Ln 31 'how they are responded'? Please review all of Box 1 for errors.

3. Table 1 is labelled participant demographics but has non demographic data such as mean size of practice. Consider revising title of table. Also, please give all details in table e.g. sex (add in number of male participants)

Results

1. P6 Ln 40-57: I'm not sure how much these themes tell us above what has already been reported in the literature on burn out, mental health and invulnerability. I would consider revising to at least incorporate what GPs attribute these experiences of mental illness and stress to.

Discussion

Overall, there is a lot of repetition in the discussion. In particular, there is constant repetition of the main findings e.g. describing the mental health problems of GPs. Discussion should be revised to avoid unnecessary repetition.

1. P10 Ln31-33: I don't think results presented support this statement. As you report, the sample was self-selected and although

	those in the 'no mental illness' group also talk about psychological stress I don't think these findings can be generalised back to the GP population as a whole. 2. P10 In 38: the authors state some GPs contemplated suicide on In 38 but at In 35 they state that almost half the sample articulated thoughts of suicide. This needs clarifying. 3. P11 In56 P12 In3-4: The authors state that some participants had taken time off work and faced barriers in seeking help and reference that this is reported in a separate paper. If the results are not presented in this analysis I think this needs to be removed. 4. P11 In 7-18: In the first sentence of this paragraph the authors state that the study adds to the literature that examines the causes and impact of chronic stress and distress in GPs yet it only describes experiences of stress and distress and does not describe any of the causes of stress. Sentence should be revised. 5. P11 In23-27: The authors state that there is evidence that doctors normalise and minimise this experience. Is there any evidence of this in their data? Can the authors add this to the interpretation of their results? I would think that this could enrich the findings. 6. P11 In35-54: The authors discuss prevention and provision of support for GPs experiencing stress and mental illness at the institutional level e.g. the GP health service and organisational support from managers and colleagues. However, there is no mention of structural level issues which the authors point out in the introduction e.g. workforce shortages and budget constraints. These are obviously harder to tackle but I think need to be mentioned in the discussion. This could be inserted in this paragraph or the following one. Conclusion Authors conclude that greater recognition needs to be given to the effects of occupational stressors. According to their topic guide they have data on this but do not present it. I think that the paper would benefit from incorporating this data and revising the objectives of the paper to do so.
--	--

REVIEWER	Josephine Stanton ADHB New Zealand
REVIEW RETURNED	23-Aug-2017

GENERAL COMMENTS	This is a large qualitative study in an important area I would think/hope there is a lot more depth in the transcript than are represented in this paper. What the paper is description of is some negative experiences described by GP's. This is part only of describing "living and working with" mental illness and distress. There was a stratified sample, some with clear mental illness, I understand, and others without experience of mental illness. But the indication seems to be that there was not a difference in their descriptions. Is this right? If so that is of interest. If no or very few doctors could be recruited in group 4 that is of interest as well. Also of interest would be how they
--

	work with these experiences, which, I would hope was asked about and described. I think there is more in the literature about experiences of doctors struggling than is cited. A particular paper which comes to mind as a qualitative study of GP's is the Thompson et al paper cited here. 1. Thompson WT, Cupples ME, Sibbett CH, et al. Challenge of culture, conscience, and contract to general practitioners' care of their own health: qualitative study. BMJ 2001;323:728–31. I think it is in the literature that doctors have horrible experiences of struggling. What this study potentially offers is a large qualitative study with many participants to give more depth to our understanding of this. More thoughtful analysis is needed to do this. One of the problems in this literature about doctors' health is the lack of differentiation between what is mental health, distress, burnout, overwork, etc. The division of the sample into different groups had the potential to address some of this but not used. The writing up needs work. There is material in the abstract which is not in the main paper, eg about groups. There seems to be slippage between qualitative and quantitative information. Eg first sentence of discussion talking about a significant population of GP's. The paper reports experiences. 47 is a lot for a qualitative study but tells us nothing about a population. Again in line 47 p 10, second paragraph of discussion "many GP's". I think it is true, but not shown by this paper. P 11 line 17-18 they talk of the group without reported problems increasing generalisability. This is conceptually messy. As I said above the differences in this group, present or not are not made clear. Generalisability is not a particularly useful concept here with a self selected group. The discussion goes way beyond what is reported in the results. I suspect there is a lot more depth in the interviews which may contribute to discussion but not presented.
--	---

VERSION 1 – AUTHOR RESPONSE

Reviewer: 1

Reviewer Name

Dr Billy Gazard (on behalf of Dr Stephani Hatch)

Institution and Country

King's College London

Recommendation: Major revisions

Reviewer's comments:

Summary: This paper aims to explore GP experiences of living and working with chronic stress, mental ill health and psychological distress using qualitative research methods. The methods are well described and researchers have clearly demonstrated the rigour and validity of the study design and procedures.

However, I have concerns about the novelty and implications of the objectives and results of the study given findings already present in the literature that not only describe the extent of psychological distress in doctors and general practitioners but also further describe the relationship between work related risk factors, burn out and psychological distress. The authors point out that few studies have explored these experiences using qualitative research methods and the descriptions that they present are useful in detailing the psychological distress GPs experience. However, I don't think the results or interpretation of the findings in this study build on the quantitative data presented in the research literature. From the presented topic guide, the authors show that they have collected information on both work-related stress and managing stress.

Author's Response:

We agree that there is a literature on this topic area, yet most of this is quantitative.

This paper is one of a set of three papers presenting the results of our analysis of this large data set. Because of the large amount of rich data obtained we have presented the results in the following papers:

Spiers, J., Buszewicz, M., Chew-Graham, C., Gerada, C., Kessler, D., Leggett, N., Manning, C., Taylor, A., Thornton, G., and Riley, R. (2017). What are the barriers, facilitators and survival strategies for GPs seeking treatment for distress? A qualitative study. BJGP. <https://doi.org/10.3399/bjgp17X692573>.

This paper reports the following findings: (1) Barriers and facilitators to help seeking for distress - work related, symptom related, and access to support (2) Survival strategies - cutting down or varying workload and asserting boundaries.

Riley, R., Spiers, J., Buszewicz, M., Taylor, A.K., Thornton, G., Chew-Graham, C. (2017). What are the Sources of Stress and Distress for General Practitioners Working in England? A Qualitative Study. BMJ Open (in Press).

This paper reports the key sources of stress/distress related to: 1) Emotion work - the work invested and required in managing and responding to the psychosocial component of GPs' work, and dealing with abusive or confrontational patients; 2) Practice culture - practice dynamics and collegial conflict, bullying, isolation and lack of support; 3) Work role and demands - fear of making mistakes, complaints and inquests, revalidation, appraisal, inspections and financial worries.

We have referred to these papers and the content covered in the Introduction (p.4/5), Discussion (p. 11) and Conclusion (p.12) of this current paper under review, so hope it is clear how these three papers fit together.

In this current paper, we aim to provide a detailed insight into the experiences of GPs living and working with chronic stress, burnout, distress or a mental health condition by offering clear descriptions of the intensity of their symptoms and distress. To the best of our knowledge, no study has provided in-depth accounts of the thoughts and feelings of this population of doctors, and in particular the frequency of their marked suicidal ideation. Most work on suicidality is quantitative and, sadly, retrospective and based on mortality rates. Our paper is an experiential account of GP participant's experiences with rich text to support the findings. The survey data typical of quantitative studies highlights the extent of the stress and burnout affecting this population, but cannot provide such rich experiential data derived from using in-depth interviews, as this study provides.

Reviewer's comments:

I would question why they have not incorporated this into the paper especially as the authors allude to these factors in their strengths and limitations section, introduction and discussion. I would recommend major revisions to fully incorporate these into the paper by updating the objectives of the study. Specific suggestions are outlined below:

Abstract

1. Page 2 Ln 54-55: In objectives, I think it's confusing for the reader that the wider study is mentioned. I think it's important to mention this in the methods but not sure it's well paced here.

Author's Response:

Where one or more papers (detailed above) are derived from a wider study, we feel it is important to link these in order to inform the reader and to provide context.

Reviewer's comments:

2. Page 3 Ln 26-32: In conclusions, I'm not sure that the author can state that there is a considerable degree of mental ill health among GPs due to the self-selecting nature of the sample. I agree with the need for greater recognition of the effect of occupational stress and don't see why this data is not incorporated into the paper if the authors have this data.

Author's Response: We have now qualified this statement to:

'These findings paint a concerning picture of the situation affecting primary care doctors, with participants' accounts suggesting there is a population of GPs experiencing a considerable degree of mental ill health and reduced wellbeing amongst GPs.'

The second BMJ Open paper (Riley et al. 2017) paper, referred to above, reports findings related to the causes and sources of stress and distress for GP participants; we agree that this was not clear and we thank the reviewer for this helpful comment.

Reviewer's comments:

1. Strengths and limitations

Page 3 Ln 46: Authors state that the study contributes to body of research on the causes and impact of chronic stress but do not present any findings on the causes of the stress and psychological distress.

Author's Response: The findings related to the causes of stress are reported in a separate paper which has been accepted for publication in BMJ Open (in press).

Reviewer's comments:

2. Page 3 Ln 49-52: authors state a strength of the study is that the sample was varied by socio-demographics but do not go on to state/describe if there were any differences in experiences by key demographics. Are there any differences?

Author's Response: In terms of participants' symptomology and experience of distress, there were no demographic differences across the groups. We have amended this section to highlight this (p.11): However, despite this heterogeneity within the sample, we did not observe any marked differences between the groups, suggesting a certain uniformity of distress for the group despite demographic differences.

Reviewer's comments:

Introduction

1. P4/5 Ln 48-57/ Ln 3-6: The second paragraph of the introduction contains important information on the risk factors for poor mental health in the GP population and how these risk factors are becoming increasingly burdensome. However, I'm not sure why this is presented given the stated objective. It

would be useful to see this literature as a basis for a second objective and findings related to this relationship presented in the results section.

Author's Response: We feel that referencing research highlighting such risk factors contributing to the extent of ill health amongst this population provides important context for this study in terms of indicating the potential relationship between the sources of stress and its effect/impact on the individual. The related BMJ Open paper (Riley et al. 2017, in press), referenced above and included in the Background section, provides a more detailed background literature on the sources of stress for GPs and provides a clearer thus context for our findings related to the impact of the sources of stress/distress for GPs participants reported in this paper. However, we do take the reviewer's point that this literature is more relevant to our findings around the causes of distress for GPs, which are discussed in our in press BMJ Open paper. We have therefore shortened this paragraph and made its relevance clearer; see pages 4-5.

Reviewer's comments:

2. P5 In 7-13: This paragraph states that few qualitative studies have explored GPs experience of working with stress and mental ill health and this seems to be the reason for conducting the analysis. This doesn't convince me as to why the analysis should be conducted given that we know the extent of mental ill health and burnout in GPs using quantitative data. It should be clearer in what this study adds to the literature. Are you aiming to provide more detailed information on the character of stress and mental ill health in GPs than what has been previously reported in the literature? A better argument for the objective of your study would be beneficial.

Author's Response:

In our introduction we are clear that quantitative data (e.g. via surveys or validated instruments) have revealed the extent of these issues amongst the GP population and we justified the value of conducting a larger scale qualitative study on the basis that it provided GP participants with an opportunity to voice the full extent and complexity of their distress and mental ill health in order to provide a more complete and detailed picture of their experiences (qualified in the study objective). Employing a qualitative study to explore this topic also provides further context to GP's experiences in understanding how and why they may feel so distressed, in addition to the sources of stress/distress for GPs (see Riley et al. 2017 described in the paired BMJ Open paper, in press) and what helps and hinders them when they seek support (see Spiers et al, 2017 in the BJGP).

Reviewer's comments:

Methods

1. In Box 1, topic point on describing average working day appears in both introduction and background and current wellbeing. One needs to be deleted.

Author's Response: We have now deleted this.

Reviewer's comments:

2. In Box 1, there are several grammatical errors, for example, on In 31 'how they are responded'? Please review all of Box 1 for errors.

Author's Response: Thank you for this comment. We have now edited Box 1.

Reviewer's comments:

3. Table 1 is labelled participant demographics but has non demographic data such as mean size of practice. Consider revising title of table. Also, please give all details in table e.g. sex (add in number of male participants)

Author's Response: Thank you for this comment. We have now changed the title of Table 1 to: Participant and practice characteristics (p.16).

Reviewer's comments:

Results

1. P6 In 40-57: I'm not sure how much these themes tell us above what has already been reported in the literature on burn out, mental health and invulnerability. I would consider revising to at least incorporate what GPs attribute these experiences of mental illness and stress to.

Author's Response: As previously described, the sources of stress experienced by these GPs have been fully reported in a linked paper, accepted by BMJOpen – and they will be presented as a pair of papers. We are now able to include the reference for this paper and have made reference to this in the text of this current paper (p.4,5 11, & 12).

Reviewer's comments:

Discussion

Overall, there is a lot of repetition in the discussion. In particular, there is constant repetition of the main findings e.g. describing the mental health problems of GPs. Discussion should be revised to avoid unnecessary repetition.

Author's Response: We thank the reviewer for this comment and we have now edited this section and removed any repetitions.

Reviewer's comments:

1. P10 In31-33: I don't think results presented support this statement. As you report, the sample was self-selected and although those in the 'no mental illness' group also talk about psychological stress I don't think these findings can be generalised back to the GP population as a whole.

Author's Response: We have replaced the term generalizable with transferable.

Reviewer's comments:

2. P10 In 38: the authors state some GPs contemplated suicide on In 38 but at In 35 they state that almost half the sample articulated thoughts of suicide. This needs clarifying.

Author's Response: We agree and have changed some to many.

Reviewer's comments:

3. P11 In56 P12 In3-4: The authors state that some participants had taken time off work and faced barriers in seeking help and reference that this is reported in a separate paper. If the results are not presented in this analysis I think this needs to be removed.

Author's Response: We have now removed this sentence.

Reviewer's comments:

4. P11 In 7-18: In the first sentence of this paragraph the authors state that the study adds to the literature that examines the causes and impact of chronic stress and distress in GPs yet it only describes experiences of stress and distress and does not describe any of the causes of stress. Sentence should be revised.

Author's Response: We have modified this sentence to:

The findings from this study and related papers (Spiers et al. 2017; Riley et al. 2017) were based on in-depth interviews with 47 GPs from across England and contribute to a growing corpus of research illuminating and examining the causes, experiences and impact of chronic stress and distress amongst GPs.

Reviewer's comments:

5. P11 ln23-27: The authors state that there is evidence that doctors normalise and minimise this experience. Is there any evidence of this in their data? Can the authors add this to the interpretation of their results? I would think that this could enrich the findings.

Author's Response:

We agree and we have now added in the following to give examples of the way in which participants normalise symptoms:

Strikingly, one GP participant (P11) normalised the way they routinely ruminated on their suicidal thoughts: "And um constant ruminations on suicide and doing myself in. And the trick with suicide is to persuade yourself that those around you would be better off without you, um and I got to that point and I decided on ways that I might (pause) do myself in and ruminated about that all the time. Just, it just filled my waking and sleeping nights." (P20, Male Partner).

Other participants indicated that they were living and coping on a day-to-day basis with symptoms of chronic stress, anxiety and depression, ranging from sleeplessness to persistent anxiety. This is discussed in the Discussion on page 11.

Reviewer's comments:

6. P11 ln35-54: The authors discuss prevention and provision of support for GPs experiencing stress and mental illness at the institutional level e.g. the GP health service and organisational support from managers and colleagues. However, there is no mention of structural level issues which the authors point out in the introduction e.g. workforce shortages and budget constraints. These are obviously harder to tackle but I think need to be mentioned in the discussion. This could be inserted in this paragraph or the following one.

Author's Response:

Thank you for this comment. We have amended the following sentence in the conclusion to include this:

The solutions are complex and lie in addressing the systemic causes of stress (related to diminishing resources, workforce shortages). The solutions relating to the systemic causes of stress are discussed in more depth in the paired paper, Riley et al. 2017 (BMJ Open in press).

We have also included the following addition to highlight the culture change as an additional factor in prevention and where this needs to be tackled: Medical schools, medical institutions and individuals need to address and tackle the stigma of mental illness and engender a greater tolerance and acceptance of vulnerability (p13).

Reviewer's comments:

Conclusion

Authors conclude that greater recognition needs to be given to the effects of occupational stressors. According to their topic guide they have data on this but do not present it. I think that the paper would benefit from incorporating this data and revising the objectives of the paper to do so.

Author's Response: This data has been reported in the paired paper Riley et al 2017 and which is also referred to in the Introductory section of this paper.

Reviewer: 2

Reviewer Name

Josephine Stanton

Institution and Country
ADHB
New Zealand

Reviewer's comments:

This is a large qualitative study in an important area

I would think/hope there is a lot more depth in the transcript than are represented in this paper. What the paper is a description of some negative experiences described by GPs. This is part only of describing "living and working with" mental illness and distress. There was a stratified sample, some with clear mental illness, I understand, and others without experience of mental illness. But the indication seems to be that there was not a difference in their descriptions. Is this right? If so that is of interest. If no or very few doctors could be recruited in group 4 that is of interest as well. Also of interest would be how they work with these experiences, which, I would hope was asked about and described.

Author's Response: We thank you for your comments. We agree that this study yielded rich and detailed descriptions which are reported in two additional papers which we are now able to reference (see above). We agree that this was not clear from the outset. The first paper details the barriers and facilitators to help-seeking while the second paper examines the sources of stress/distress for GPs and is paired with this paper, which provides a detailed account of the experiences of GPs living and working with stress/distress and symptoms of mental ill health and the impact of the sources of stress detailed in the paired paper. The references for this paper are detailed above.

Reviewer's comments:

I think there is more in the literature about experiences of doctors struggling than is cited. A particular paper which comes to mind is a qualitative study of GP's is the Thompson et al paper cited here. Thompson WT, Cupples ME, Sibbett CH, et al. Challenge of culture, conscience, and contract to general practitioners' care of their own health: qualitative study. *BMJ* 2001;323:728-31.

Author's Response: Thank you for directing us to this reference, which we have now cited in the Introduction. We agree that our findings are supported by this literature but also build on it by reporting the ways in which distress is experienced by GPs. Specifically, GP participants in this study gave very clear and moving accounts of having suicidal thoughts, which are under reported in the literature. We have added a new sub-theme (p.8) under Depression titled: Distressed at work which reveals how participants' symptomology and levels of distress manifest at work (see below). This is also underreported in the literature. Most evidence on suicide is quantitative and retrospective, reporting mortality amongst doctors. The Thompson et al. paper does not include reports of suicidality amongst their participants, and therefore our findings are novel in this sense.

Distressed at work:

Many participants recalled their experiences of feeling distressed, anxious and tearful, with some describing breaking down at home, after or during work or in front of patients:

I mean there were times certainly when I arrived at work and just sat outside and cried before I even went in the building. And I often cried on the way home. I'd often cry when I came in from work, um just collapse in a heap, and get to bed actually crying. (P31, Female Locum)

I remember once actually sitting under the desk, hugging my knees to my chest, I was so anxious. (P25, Male Salaried)

I would just end up in tears all the time at the end of each session. (P53, Female Salaried)

We have also supplemented the theme related Shame and feelings of failure (p.10) with the following quote in order to highlight the consequences of internalised stigma:

'Yeah, well you don't get ill, do you? You know, you, you refuse to accept any illness in yourself, don't you? It's just a question of that's why doctors present late with illnesses because they don't like to make a fuss. We see it as a sign of weakness, which I think probably is why we are intolerant of illness in our partners, because we're doctors. And I think it's just part of the sort of macho thing which is first drummed into you at medical school and the end product of that generation, which – if you can't take the heat get out of the kitchen, type thing'. (P49, Female, Retired)

Reviewer's comments:

I think it is in the literature that doctors have horrible experiences of struggling. What this study potentially offers is a large qualitative study with many participants to give more depth to our understanding of this. More thoughtful analysis is needed to do this. One of the problems in this literature about doctors' health is the lack of differentiation between what is mental health, distress, burnout, overwork, etc. The division of the sample into different groups had the potential to address some of this but not used.

Author's Response:

Participants were sampled on the basis of their experience of help-seeking, since the overarching study aim was to explore barriers and facilitators to help-seeking amongst GPs. These barriers and facilitators have been published separately (Spiers et al. 2017 BJGP), and the findings apply across the groups. The topic guide and interviews made provision for GPs to provide their personal meaning of mental ill health, and interviewees responded to questions about why participants sought help. We are therefore reporting participants' descriptions/accounts to highlight the range of symptomology amongst the population, including amongst those with no self-defined illness or who had never sought help. The study team wished to avoid making any formal assessment of participants' symptomology and relied on verbatim accounts and grouped those accordingly. The exception to this was when participants employed diagnostic labelling which was made clear in the text (p.5). There was no clear distinction in terms of presentation or severity of symptoms between the groups and so no typology could be developed along these lines. We have included a sentence pertaining to this in the text (p.7).

Reviewer's comments:

The writing up needs work. There is material in the abstract which is not in the main paper, eg about groups. There seems to be slippage between qualitative and quantitative information. Eg first sentence of discussion talking about a significant population of GP's. The paper reports experiences. 47 is a lot for a qualitative study but tells us nothing about a population. Again in line 47 p 10, second paragraph of discussion "many GP's". I think it is true, but not shown by this paper.

Author's Response:

Thank you for your comment – in clarification of the points raised:

- A further description of groups 1-4 is included in para 1 of the methods section, on page 5.
- A significant population of GPs has been changed to undetermined in the first line of the Discussion.
- All other qualifiers (i.e. many, some) on page 10 refer to the study population and are reflected in the frequency with which themes emerged in participants across the data.

Reviewer's comments:

P 11 line 17-18 they talk of the group without reported problems increasing generalisability. This is conceptually messy. As I said above the differences in this group, present or not are not made clear. Generalisability is not a particularly useful concept here with a self-selected group.
Author's Response: We have replaced the term generalizability with transferability.

Reviewer's comments:

The discussion goes way beyond what is reported in the results. I suspect there is a lot more depth in the interviews which may contribute to discussion but not presented.
Author's Response: The discussion now includes references to our other papers, which provide further context and detail to the findings reported in this paper. The discussion has incorporated relevant evidence which related to our study findings.

Many thanks again for all your thoughtful comments. We hope that you agree that this paper is now ready for publication in the BMJ Open.

Ruth Riley and co-authors

VERSION 2 – REVIEW

REVIEWER	Dr Billy Gazard (on behalf of Dr Stephani Hatch) King's College London
REVIEW RETURNED	01-Nov-2017

GENERAL COMMENTS	The authors have addressed my comments fully. I think the paper has really benefited from further details on how the objective of this study sit with the objectives of the larger project. The added references related to the other two papers published from this project also make things much clearer for the reader.
--

REVIEWER	Josephine Stanton Auckland Healthcare New Zealand
REVIEW RETURNED	15-Oct-2017

GENERAL COMMENTS	I am sorry but I still feel as if the comments I made last time are not addressed. Other qualitative studies are cited, but their results not brought forward in the introduction or the discussion to look at how the results of this study mesh with others. The discussion is not predicated on the results. It does not use the "in depth" data offered by the study. It says a lot of sensible things but these could just as well be based on survey and other data already in the literature. I think the authors should ask themselves what there is in the discussion they would say after having done this study that they would not have had the data to say before they did the study. Also, if the group self selecting as not having mental health issues is no different from the others this is very important. Were almost half of them describing suicidal thoughts? There is very little in the results about how the doctors manage their struggles, what gets them through. This is a self selecting group, with or without mental health issues. This is valuable for bringing out the texture, detail and nature of experiences, but generalizability is not the appropriate measure.
--

VERSION 2 – AUTHOR RESPONSE

Response to reviewer's comments #2

Response to reviewer 1

The authors are grateful for your previous review of our paper and we are pleased that the paper has benefited from addressing your comments.

Response to reviewer 2

I am sorry but I still feel as if the comments I made last time are not addressed. Other qualitative studies are cited, but their results not brought forward in the introduction or the discussion to look at how the results of this study mesh with others.

Response: We thank the reviewer for this comment. The literature included in the background section provides a context for the mental ill health and includes key studies on the mental ill health of doctors, some of which are reflected in the discussion. The literature on suicidality amongst general practitioners is limited to quantitative studies, many of which are out of date. This therefore provides an important context for our work.

The discussion is not predicated on the results.

We thank the reviewer for this comment but the authors do not agree with this point. The discussion section refers to the key findings, focusing on those which are underreported in the literature such as suicidality and distress in the workplace. The discussion incorporates relevant literature in terms of author suggestions for solutions in terms of preventing mental ill health amongst doctors such as interventions to reduce stress/distress and the importance of tackling socio-cultural barriers in medicine which prevent doctors from seeking help.

It does not use the "in depth" data offered by the study. It says a lot of sensible things but these could just as well be based on survey and other data already in the literature. I think the authors should ask themselves what there is in the discussion they would say after having done this study that they would not have had the data to say before they did the study.

We thank the reviewer for this comment. The authors feel that this paper and related published papers have endeavoured to report the depth and breadth of rich data which the study participants have generated in this large qualitative study. The range of symptomology articulated by the study participants is reflected in these findings and contributes insight into what GPs may be feeling and experiencing while communicating accounts of GPs who are distressed yet continue to work. The accounts of suicidal thoughts are novel as are underreported in the literature. We were struck by the levels of distress and significant physical and psychological symptoms articulated by many of our participants and are reflected in the selected quotes and Discussion. We are not aware of any previous publications describing this in such vivid detail. We have included a sentence to highlight these novel findings in the Discussion on p.11

Also, if the group is self selecting as not having mental health issues is no different from the others this is very important. Were almost half of them describing suicidal thoughts?

Thank you for asking for clarification on this point. We have carefully checked through our raw data on NVivo and found that 13/47, or just over one in four participants, expressed suicidal thoughts. No group four participants articulated thoughts of suicide but some made reference to witnessing the suicides of colleagues. We have updated the manuscript to address this (pages 9-10).

Comment: There is very little in the results about how the doctors manage their struggles, what gets them through.

Response: As the reviewer suggests, study participants did talk about their potential coping strategies. These are reported and discussed elsewhere in the Spiers et al (2017) paper, but was not the aim of this current manuscript.

Comment: This is a self selecting group, with or without mental health issues. This is valuable for bringing out the texture, detail and nature of experiences, but generalizability is not the appropriate measure.

Response: We thank the reviewer for this comment and we agree – we're not suggesting generalizability and employ the term 'transferability' which is a recognised term in qualitative research (Lincoln and Guba, 1985)